# Health Impact of Air Pollution from Shipping in the Baltic Sea: Effects of Different Spatial Resolutions in Sweden

**DOI:** 10.3390/ijerph17217963

**Published:** 2020-10-29

**Authors:** Nandi S. Mwase, Alicia Ekström, Jan Eiof Jonson, Erik Svensson, Jukka-Pekka Jalkanen, Janine Wichmann, Peter Molnár, Leo Stockfelt

**Affiliations:** 1School of Health Systems and Public Health, Faculty of Health Sciences, University of Pretoria, Pretoria 0002, South Africa; janine.wichmann@up.ac.za; 2Occupational and Environmental Medicine, School of Public Health and Community Medicine, Institute of Medicine, University of Gothenburg, Box 100, 405 30 Gothenburg, Sweden; Alicia.Ekstrom@hotmail.se (A.E.); peter.molnar@amm.gu.se (P.M.); 3Division for Climate Modelling and Air Pollution, Norwegian Meteorological Institute, NO 0313 Oslo, Norway; janeij@met.no; 4The Environmental Department, City of Gothenburg, PO Box 7012, 402 31 Gothenburg, Sweden; erik.svensson@miljo.goteborg.se; 5Air Quality Research, Finnish Meteorological Institute, PL 503, FI-00101 Helsinki, Finland; jukka-pekka.jalkanen@fmi.fi; 6Department of Infection Diseases, Sahlgrenska Academy, University of Gothenburg, SE-405 30 Goteborg, Sweden; leo.stockfelt@amm.gu.se

**Keywords:** PM_2.5_, SECA, EMEP model, population exposure, health effects, myocardial infarction, heart attack, stroke, air pollutants

## Abstract

In 2015, stricter regulations to reduce sulfur dioxide emissions and particulate air pollution from shipping were implemented in the Baltic Sea. We investigated the effects on population exposure to particles <2.5 µm (PM_2.5_) from shipping and estimated related morbidity and mortality in Sweden’s 21 counties at different spatial resolutions. We used a regional model to estimate exposure in Sweden and a city-scale model for Gothenburg. Effects of PM_2.5_ exposure on total mortality, ischemic heart disease, and stroke were estimated using exposure–response functions from the literature and combining them into disability-adjusted life years (DALYS). PM_2.5_ exposure from shipping in Gothenburg decreased by 7% (1.6 to 1.5 µg/m^3^) using the city-scale model, and 35% (0.5 to 0.3 µg/m^3^) using the regional model. Different population resolutions had no effects on population exposures. In the city-scale model, annual premature deaths due to shipping PM_2.5_ dropped from 97 with the high-sulfur scenario to 90 in the low-sulfur scenario, and in the regional model from 32 to 21. In Sweden, DALYs lost due to PM_2.5_ from Baltic Sea shipping decreased from approximately 5700 to 4200. In conclusion, sulfur emission restrictions for shipping had positive effects on health, but the model resolution affects estimations.

## 1. Introduction

Air pollution is one of the world’s most detrimental environmental health risks, with outdoor and indoor air pollution together annually accounting for 4.1 million premature deaths worldwide [1]. Many studies have shown that sulfur dioxide (SO_2_), nitrogen oxides (NO, NO_2_), and particulate matter (particles <2.5 µm (PM_2.5_) and particles <10 µm (PM_10_)) have strong associations with adverse cardiopulmonary health effects [2,3,4]. Long-term exposure to PM_2.5_ is highly associated with adverse health effects such as ischemic heart disease, myocardial infarction (MI), and stroke, and mortality [3,5,6,7,8,9,10,11,12]. The multiple sources of air pollution include anthropogenic and natural emissions [13]. One anthropogenic source is shipping emissions, contributing an estimated 13% of global sulfur emissions [14]. PM2.5 from shipping consists of both primary emissions and secondary particles from SO2 [15]. Sulfur in the form of SO_2_ is one of the key precursors for particulate matter (PM) [16,17], and the reduction in sulfur content in shipping fuel impacts the presence of both PM_2.5_ and PM_10_. 

Countries across the world have designated sulfur emission control areas (SECAs) with governance from the International Maritime Organization (IMO) [18,19,20]. Studies have shown that shipping emissions increase the presence of SO_2_ and PM in harbor cities [21,22]. The SECAs stipulated the areas subject to guidelines to reduce the sulfur content in shipping fuel. The gradually tightening marine fuel sulfur (S) regulations (maximum allowed fuel sulfur content decreased from 3.5% to 1.5% in 2006 and to 1.0% in 2010) have significantly decreased the SO_x_ and PM emissions from Baltic Sea shipping and from 2015 onwards, only 0.1%S fuel is allowed in the area. The 2015 change only concerns regional sulfur rules though since low-sulfur fuel has been mandatory in port areas in the EU since 2010. In addition to regional sulfur limits, low-sulfur fuel must be used in EU port areas from 2010 onwards as required by the EU Sulfur Directive, [23,24]. Globally, outside the SECAs (Figure 1), the maximum allowed fuel sulfur content was decreased from 3.5% to 0.5% on 1 January 2020 [24]. 

In recent decades, the concentrations of air pollutants have decreased in Northern Europe and Sweden [16,21,25,26]. The decrease in particle emissions from shipping due to stricter SECA regulations in the Baltic Sea and the North Sea has been estimated to have led to substantial health improvements, and further restriction will lead to additional benefits [23,27]. 

Recent studies investigating PM_2.5_ exposure levels in Sweden and other European countries [28,29] have used only coarse resolutions to assess population exposures. Few studies have used finer population grids or assessed how such refinement may affect the assumptions made about different health outcomes [25]. Therefore, the aim of this study was to investigate how mortality and morbidity rates associated with PM_2.5_ exposure differed depending on spatial resolutions in Gothenburg, Sweden, before and after stricter sulfur regulations were implemented in 2015. We also examined the health impact in Sweden and its 21 counties before and after sulfur regulations.

## 2. Materials and Methods

### 2.1. Modeling and Geocoding

In this paper, three exposure models were used. First, the Ship Traffic Emission Assessment Model (STEAM; [30,31,32]) was used to generate ship emission inventories for consecutive chemical transport modeling steps. STEAM takes as input ship position data from the Automatic Identification System (AIS), which enables tracking of individual vessels with very high spatio-temporal resolution (meters, seconds). These activity data were provided by the Baltic Sea (Helsinki Commission) countries. STEAM models vessel water resistance, power use, and emissions based on technical description of ships [33]. For this work, both regional and local (city-scale, near harbors) emission inventories for ship emissions were generated for 2014, 2015, and 2016 with STEAM. Appendix A shows the summary of ship emissions for both the Baltic Sea and the smaller domain around the Gothenburg area. Appendix A shows the SO_x_ emissions (kg/cell) from the Baltic Sea in 2016 and Appendix A shows the difference plot in 2014–2016, around Gothenburg, and illustrates the major shipping lanes. 

Further, we have used two chemical transport models (CTM) to assess both total PM_2.5_ air concentration and the contribution from shipping to PM_2.5_ in the Baltic Sea. The first CTM, which we will refer to as the regional model, was that of the European Monitoring Evaluation Programme (EMEP), which modeled Europe, including Sweden, in 0.1 × 0.1 (approximately 10 × 10 km) grid squares [34]. The EMEP model has been described in detail previously [35]. The STEAM data were used as input to the EMEP model. The EMEP model was run including all emissions, with high sulfur ship emissions, low sulfur ship emission, and with no ship emissions for the years 2014, 2015, and 2016. The differences “high sulfur - no shipping” and “low sulfur – no shipping” for each year give the shipping contribution with high- and low-sulfur fuel. By calculating the average of the three years for high- and low-sulfur fuel, respectively, the influence of different meteorology between the years was evened out. These models and scenarios have been described in more detail previously [34]. The resulting three-year average model results are named high-sulfur and low-sulfur scenarios throughout the paper. Additionally, we used the population in Sweden for 2015 for all exposure assignments to avoid an effect of possible population fluctuations. 

The third model, which we will refer to as the city-scale model is run within the Airviro system [36], an air pollution model system widely used [37,38,39]. Airviro uses the Danard calculations within the Airviro Gauss module (“meteorological pre-processor”). We feed Airviro Gauss with topography, physiography, and weather station data and the module calculates an hour-by-hour wind field over the chosen area with a grid size of 100 × 100 m for the city of Gothenburg, Sweden. The local environmental office in Gothenburg calculated PM_2.5_ concentrations in this city-scale model with meteorology from 2016 using two scenarios, one with and one without the 2015 SECA enforcement, also in this paper referred to as high-sulfur and low-sulfur scenarios. All land-based emissions and in-harbor area ship emissions in both scenarios were from the emission inventory for the year 2016. The regional EMEP model was used as an input in the city-scale model’s boundary as a transport source, but within the city-scale’s model domain, the regional model grid sources were not used. STEAM emissions were used in both CTMs, but with a different higher-resolution input for the Airviro model. Identical methods for ship emission inventories were used, but the air quality modeling uses EMEP for regional and Airviro for local study. The merging of the concentration fields from the different models was done by projecting the regional model results on to the 100 × 100 m grid of the local model using linear interpolation. The concentration field from the local modeling was then added. The results from this procedure corresponded well with PM_2.5_ measurements made by the local environmental office in Gothenburg (84–114% of mean annual measurements from permanent stations) [40], without any empirical adjustment of the model results being made.

Data from Eurostat [41] were used to geocode the population of Sweden into 1 × 1 km squares. We also investigated two other resolutions of the spatial distribution of the population in the local Gothenburg analysis. These population datasets included population centroids [42] and the Swedish national population grid (100 m grid size) from Statistics Sweden [43].

A geographical information system in the QGIS program, version 2.18.13, was used in the geocoding and to link air pollution data to inhabitant data. Fixed population data for the year 2015 were used in all calculations to avoid fluctuations due to population changes.

### 2.2. Baseline Population

In 2015, Sweden had a population of approximately 9.5 million registered residents in 21 counties. Stockholm was the largest county, with a population over 2.1 million inhabitants, whereas Gotland County had the fewest inhabitants (<60,000; Table 1). The population of Gothenburg, the largest city in Västra Götaland County, was approximately 650,000 in 2015.

### 2.3. Exposure Assessment

We calculated the population-weighted exposure to PM_2.5_ by combining the regional model EMEP and the Eurostat data. All inhabitants were assigned exposure to PM_2.5_ in each 1-km square in both high- and low-sulfur scenarios for each of the 21 counties (Table 1), and the population-weighted means were calculated. For the Gothenburg analysis, we compared the EMEP model with the city-scale model using both high- and low-sulfur scenarios for the three population grids.

### 2.4. Exposure–Response Function and Mortality Calculation

Two linear exposure–response (ER) functions were used for all-cause mortality from long-term exposure to PM_2.5_. The first is from the European Study of Cohorts for Air Pollution Effects (ESCAPE) project with a relative risk of 1.07 (95% CI 1.02–1.13) per 5 µg/m^3^ increase in PM_2.5_ for all-cause mortality [12]. The second ER function was derived from the Health Risks of Air Pollution in Europe (HRAPIE), with a relative risk of 1.062 (95% CI 1.04–1.08) per 10 µg/m^3^ increase in PM_2.5_ for all-cause mortality [44]. The two ER functions provide a comparative assessment of the all-cause mortality from shipping emissions. The ESCAPE ER function is more recent and estimates a 14% increased risk of all-cause mortality per 10 µg/m^3^ of PM_2.5_ exposure versus the older HRAPIE ER function of 6.2% per 10 µg/m^3^ of PM_2.5_ that has been used in most previous health impact assessments [45]. The premature deaths attributable to PM_2.5_ can be seen as those deaths that could have been avoided if PM_2.5_ concentrations from shipping had been reduced to 0 µg/m^3^ everywhere in Sweden (calculation found in the Appendix A).

There are 21 counties in Sweden and the majority of the population is located in the southern part of the country. We used the 2015 natural (non-accidental) mortality for those aged ≥30 years for each county [43].

### 2.5. Years of Life Lost (YLL) Calculations

We used data life tables from Statistics Sweden (SCB) and Eurostat [41] to calculate the YLL for all counties in Sweden and age-specific death and survival rates from the life tables to estimate the corresponding reduction in life expectancy from PM_2.5_ exposure from shipping for the average person in the Swedish population aged ≥30 years, per year.

The YLL calculation was used to determine disability-adjusted life years (DALYs) as a measure of the estimated burden of disease in Sweden, its 21 counties, and the city of Gothenburg.

### 2.6. Morbidity Calculations

MI and stroke were the health outcomes considered for the morbidity estimation, because heart disease and stroke are important health outcomes and the most common reasons for premature deaths attributed to exposure to air pollution [7,46]. The relative risk per 5 µg/m^3^ increase in PM_2.5_ used for MI was 1.13 (95% CI 0.98–1.30) [47], and for stroke it was 1.19 (95% CI 0.88–1.62) [48]. The Swedish National Board of Health and Welfare’s statistical database provided all the data regarding MI and stroke events in Sweden, its counties, and the city of Gothenburg [43]. The relative risk was calculated from the average population exposure and the number of first-time incidents for the year in question. We avoided double counting by subtracting the number of deaths caused by MI and stroke that year from the number of first-time incidents.

## 3. Results

### 3.1. Exposures in Gothenburg

The city-scale model estimated a total mean PM_2.5_ concentration of 6.92 µg/m^3^ in the high-sulfur scenario and 6.82 µg/m^3^ in the low-sulfur scenario. In comparison, the regional model recorded total mean PM_2.5_ concentration levels of approximately 4.9 µg/m^3^ in the high-sulfur scenario and 4.6 µg/m^3^ in the low-sulfur scenario. Using both models, the total PM_2.5_ concentration levels were below both the European annual air quality guideline of 25 µg/m^3^ and the World Health Organization (WHO) recommended annual average of 10 µg/m^3^ [8]. The average population exposure in Gothenburg to PM_2.5_ from Baltic shipping was approximately 1.5 µg/m^3^ in the high-sulfur scenario and 1.6 µg/m^3^ in the low-sulfur scenario for all the different population grids for the city-scale model. Using the regional model, the average population exposures for the different population grids were approximately 0.5 µg/m^3^ in the high-sulfur scenario and 0.3 µg/m^3^ in the low-sulfur scenario (Table 2). 

The mean population exposure-from-shipping difference before and after the revised SECA regulations was smaller in the city-scale model (Figure 2) than in the regional model (Figure 3). Exposure concentrations in the city-scale model show a clearer gradient from the areas closer to the shipping source as the exposure levels enter the inland areas, as opposed to the regional model (Appendix A). The choice of population dataset (data centroids, 1 km grid or 100 m grid) did not affect the mean population exposure in either the regional or city-scale models.

### 3.2. Exposure in Sweden

In Sweden as a whole, the mean population exposure levels of PM_2.5_ from Baltic shipping in low- and high-sulfur scenarios were 0.35 and 0.23 µg/m^3^, respectively. Generally, the highest exposure levels were in the southern parts of Sweden, especially in counties bordering the Baltic Sea (Figure 4). Skåne County recorded the highest contribution from shipping emissions to PM_2.5_ exposure in both the low-sulfur scenario (0.73 µg/m^3^) and the high-sulfur scenario (0.54 µg/m^3^), while Jämtland County recorded the lowest population-weighted levels in both scenarios, with 0.04 µg/m^3^ in the low-sulfur scenario and 0.02 µg/m^3^ in the high-sulfur scenario (Table 1).

The population density in Sweden varied from more than 260 inhabitants per km^2^ in Stockholm to 1 inhabitant per km^2^ in sparsely populated counties in north-western Sweden (Appendix A).

### 3.3. Mortality

Levels of exposure to PM_2.5_ from shipping in the Baltic Sea and the consequent health effects decreased in both the city-scale and the regional models after the implementation of stricter SECA regulations in 2015. In Gothenburg, the city-scale model showed estimated premature deaths due to PM_2.5_ from shipping emissions decreasing from 97 to 90, while the regional model showed a decrease from 32 to 21. In the high-resolution model, the estimated YLL due to PM_2.5_ from shipping in Gothenburg also dropped from 1294 to 1206 and the average reduction in life expectancy fell from 17 hours to 16 per person. In the regional model, YLL decreased from 433 to 283 overall and from approximately 6 hours per person to 4.

Table 3 illustrates the estimated premature deaths from exposure to PM_2.5_ from Baltic Sea shipping emissions in the high- and low-sulfur scenarios with the stricter 2015 SECA sulfur fuel regulations using ESCAPE and HRAPIE ER functions. Stockholm County, the most densely populated, experienced a 44% reduction in premature deaths due to PM_2.5_ from shipping after the sulfur regulations were implemented. Other highly populated counties such as Skåne and Västra Götaland experienced 26% and 35% reductions, respectively. In Sweden, YLL decreased from 4161 in the high-sulfur scenario to 2680 in the low-sulfur scenario, a reduction of 36%.

### 3.4. Morbidity

In Gothenburg, using the average population exposure from the city-scale model, extra MI cases due to PM_2.5_ from Baltic Sea shipping decreased from 56 in the high-sulfur scenario to 52 in the low-sulfur scenario. Using the regional model, cases of MI decreased from 19 to 12. The estimated extra stroke cases dropped from 68 to 63 cases in the city-scale model and from 23 to 15 in the regional model.

Table 4 shows the estimated excess fatal MI and stroke cases due to PM_2.5_ exposure from shipping in the Baltic for Sweden and its counties for the high- and low-sulfur scenarios. MI cases in Sweden were 184 in the high-sulfur scenario and 118 in the low-sulfur scenario; stroke cases decreased from 274 to 177. The most populated counties recorded larger reductions in both MI and stroke from the high- and low-sulfur scenarios, and Stockholm County recorded the highest reductions. Reductions in Skåne, however, with the highest recorded exposure levels in both high- and low-sulfur scenarios (Table 1), were not as high as in other counties.

Using the city-scale model, the total burden of disease due to PM_2.5_ from Baltic Sea shipping in Gothenburg decreased from approximately 1374 to 1286 DALYs. Using the regional model, DALYs decreased from 513 to 363 over the same period. In Sweden, DALYs lost due to PM_2.5_ from Baltic Sea shipping decreased from approximately 5700 to approximately 4200 (Table 4).

## 4. Discussion

The stricter regional regulations of 2015 reduced the percentage of sulfur dioxide (a precursor of PM_2.5_) [16] in shipping fuel, and thereby decreased population exposure to PM_2.5_ from shipping. The largest reductions in PM_2.5_ concentrations were in the southern parts of Sweden. Skåne recorded the highest PM_2.5_ exposure levels due to its location next to the entrance to the Baltic Sea, next to major shipping lanes, while inland counties had lower levels of exposure to shipping emissions. The results for Gothenburg show that PM_2.5_ exposure from Baltic Sea shipping emissions was markedly higher in the city-scale model than in the regional model. The decrease in exposure between the high-sulfur and low-sulfur scenarios, however, was found to be larger in the regional model. Average population-weighted exposure levels were not affected by differences in population grid size (i.e., centroids, 1 × 1 km, and 100 × 100 m). This study indicates that the reductions in exposure levels after stricter sulfur regulations for shipping fuel in the Baltic Sea region have considerably decreased adverse health effects, with reductions in mortality and burden of disease from MI and stroke. However, estimations of these health effects were highly dependent on the exposure model used.

### 4.1. Exposure Assessment

When investigating the use of different resolution models to determine shipping emission exposure levels, we focused on the city of Gothenburg located in Västra Götaland County. In the city-scale model, shipping emissions accounted for 23% of the total PM_2.5_ concentration levels, while in the regional model they accounted for only 10%. The city-scale model produced a mean estimated population exposure approximately three times higher than the regional model (Table 2). Higher-resolution models have previously been shown to result in higher mean concentrations compared to lower resolution [49]. The estimated mean PM_2.5_ population-weighted exposure level of 1.5 µg/m^3^ in the city-scale model was relatively high compared with previous studies [5,24]. Those studies, however, made predictive estimates from data obtained prior to 2015 and modeled only local shipping as a separate source, not considering long-range emissions from outside the city-scale model domain. The long-range emissions could also be contributed from the North Sea, where stricter sulfur fuel regulations were simultaneously implemented in 2015, therefore a possible source of PM_2.5_ levels within the Swedish territory.

The differences between the regional- and city-scale models were highest closer to the sea and around the harbor inlet and were lower further inland, as illustrated in Figure 2 and Figure 3, and Appendix A. This is because the city-scale model more accurately captures emissions and concentration decays from the shipping routes, which are more important near strong sources. The city-scale model also provides a better estimation of how emission levels change from the source to actual population exposure. The regional model provides only average exposure levels over large grid squares, this difference illustrates the importance of using a city-scale model close to major sources of emissions. This is a smaller problem when calculating average county exposures, since average distances from shipping routes are larger. Sofiev et al. [14] encourages the use of higher-spatial resolution models that more accurately assess proximal exposure levels to at-risk communities.

We also found that different population resolutions did not influence mean population exposures. The size of the population grid (i.e., centroid, 1 km, or 100 m) had only a negligible effect on mean population exposures.

The regional model shows a greater reduction in PM_2.5_ from shipping emissions after SECA enforcement than the city-scale model (Figure 3a,b). This is because regional model concentrations are spread out over a larger area and it is not possible to reproduce the fine structure of maximum and minimum concentrations within the city with the 10 × 10 km squares equally as well as with the 100 × 100 m squares in the city-scale model. Other reasons for the low difference in the city-scale model could be that shipping in European Union port areas has already been using 0.1% fuel since 2010 [23,24], thus only the ship emissions at open sea outside the Gothenburg harbor area were impacted by the SECA regulation in 2015, and the much lower contribution from shipping in the harbor will affect the large grids in the regional model less. In addition, the city-scale model gives a more accurate concentration gradient from the source of the pollution (i.e., shipping) to the city and its inhabitants.

Due to the sensitivity, mostly from meteorological factors, high-resolution modeling is not always preferred for exposure modeling [50,51]. A study by Lui and Zhang [50] showed little or no improvements with model resolutions finer than the meteorological input data. Additionally, some studies have found that finer-resolution models do not necessarily improve model performance and so coarse resolution can still be favored [51]. However, Schaap et al. [52] showed marked improvements for short-lived species as a function of resolution, but the improvement was shown to be limited by the resolution of the input data. Furthermore, city-scale modeling can show a clearer pollution diffusion tendency with much higher concentrations in polluted areas than in the nearby areas, whereas with a coarser resolution, the spatial discrepancies are averaged out, making the distinction in the areas more difficult to determine [53]. City-scale modeling has also been found to be preferable in determining population-weighted exposure in epidemiological studies [49].

In the southern part of Sweden, more densely populated counties experienced a moderate reduction in exposure levels due to their location by the coast. Skåne recorded the highest PM_2.5_ exposure levels in both low- and high-sulfur scenarios. The county is located at the only entry and exit point for ships to and from the inner part of the Baltic Sea. Denmark, which is similarly located, also recorded the highest PM_2.5_ mean concentration in a similar study by Barregard et al. [28]. Counties located inland, such as Norrbotten and Jämtland, had minimal impact from shipping emissions due to their distance from the source of interest.

### 4.2. Mortality

The number of premature deaths attributed to exposure to PM_2.5_ from shipping emissions in city-scale model of Gothenburg in the low-sulfur scenario was 90 and 21 in the high-sulfur scenario. The number of premature deaths recorded in this study is lower than the estimated 140 premature deaths recorded by Tang et al. [25]. That study, however, predicted mortality based on 2012 data and assessed health impacts using the ALPHA-RiskPoll methodology, which includes other pollutants (NO_2_ and O_3_) as well as PM_2.5_. In the city-scale model in the present study, YLL per person calculated lower figures in the low-sulfur scenario in Gothenburg (approximately 16 hours YLL per person) than the just over 24 hours YLL per person calculated by Tang et al. [25].

In Sweden, there were an estimated 4161 YLL in the high-sulfur scenario, which were decreased by 36% to 2680 in the low-sulfur scenario due to the reduction in PM_2.5_. Areas with a high population density experienced the most significant reductions in YLL from exposure to PM_2.5_ from shipping. These reductions were also reflected in the number of premature deaths attributed to exposure to shipping emissions in individuals aged ≥30 years. Another study using country-level data observed a similar relative reduction in YLL from 4092 in a high-sulfur scenario to 2635 in a low-sulfur scenario [28]. The slight difference could be explained by our cutoff age of 30 years versus that study’s cutoff of 25; the number of cases also differed slightly due to a change in statistics and registries.

### 4.3. Morbidity

MI and stroke were chosen as indicators of the effects of emissions from Baltic shipping on morbidity in Sweden and its counties because they are serious health outcomes with strong evidence of associations with PM_2.5_ exposure [2,46,54]. Results indicated a reduction of 36% in first-time cases of both MI and stroke, corresponding to 65 annual cases of MI and almost 100 cases of stroke from the high- to low-sulfur scenarios. These reductions, although not large considering the population of Sweden, show that the stricter sulfur regulations implemented in 2015 have contributed to a reduction in morbidity. Furthermore, the counties located in the south experienced a relatively large reduction in first-time cases of MI and stroke. It can be assumed that near major shipping lanes and ports in the more densely populated coastal regions, the health impacts of PM_2.5_ from shipping are greater and the benefits of reduced emissions are larger [14].

The burdens of disease of MI and stroke differed between the high- and regional exposure models for Gothenburg. The number of DALYs decreased by 88 in the city-scale model and 150 in the regional model. Over 5700 DALYs due to MI and stroke in Sweden were attributed to PM_2.5_ in the high-sulfur scenario. This decreased by approximately 1400 DALYs in the low-sulfur scenario after the 2015 implementation of stricter SECA regulations. Stroke contributed more to the burden of disease than MI. A reduction in stroke cases relieves the health system of high healthcare expenses [55]. By implementing stricter emission regulations, Sweden and its counties experienced a relative reduction in DALYs that shows the benefits of such regulation. This is particularly significant because Sweden’s health system is funded by taxes [56]. A reduction in healthcare costs in one area increases funds available for other sectors in the health system. The stricter SECA regulations have also resulted in better overall health outcomes as predicted by Jonson et al. [23], and Brandt et al. [27].

Other European studies have shown that shipping emissions affect mortality and morbidity in surrounding populations [12,14,28]. Air pollution from shipping, however, affects not only the countries around the Baltic region, but also, through long-range (air current) transmission of PM_2.5_, areas further away [16]. The weather patterns in Northern Europe, including Sweden, mean that people spend over 80% of their time indoors during the winter. Since infiltration of outdoor air pollution is in part prevented by the filtration of air in homes and workplaces [57], it is possible that the population would be more affected during the warmer months when people spend more time outdoors. However, studies have also indicated that outdoor exposure still infiltrates indoor microenvironments such as homes and workplaces and puts individuals at risk of exposure in spite of spending a significant amount of time indoors [58,59]. Reducing air pollution from ship emissions will not only make the warmer months safer for exposed populations, but the reduced contribution of these emissions to climate change will likely influence the atmospheric chemistry and encourage governing bodies to improve on these beneficial regulations [14,25].

This study aimed to compare and contrast high- and low-spatial resolution models to account for and provide tangible results showing the differences between the two resolutions and their effects on estimated health outcomes calculated though DALYs. We found the city-scale model to be more likely to define concentrations of population exposure accurately. Although its practicality on a larger scale may be a challenge, similar methods can be used in cities that border the harbor. Using different grids for geocoding the population did not affect exposure estimations.

## 5. Limitations and Recommendations

Limitations to this study include a possible underestimation of the health effects associated with air pollution from shipping since the EMEP scenarios only considered emission changes in the Baltic Sea. The effects of the corresponding emission reductions in the North Sea have also contributed to the reductions in PM_2.5_ levels in Sweden but are not considered here. Additionally, the health effects of air pollution may have been underestimated since we only calculated effects of PM_2.5_ and did not include the effects of other air pollutants such as NO_x_, SO_x_, PM_10_, and O_3_ in this work, and the effects of PM_2.5_ were also, therefore, overestimated. Additionally, other health effects associated with air pollution such as chronic obstructive pulmonary disease [60], lower respiratory infections, type 2 diabetes [61], lower birth weight, or short gestation [62] were not considered in this study. Despite the inclusion in the ESCAPE project of several European countries, use of the ESCAPE cohort’s RR in studies to show the association between air pollution (namely PM_2.5_) and all-cause mortality [63] has drawn some criticism. Some studies used in the ESCAPE project show a moderate association between exposure to PM_2.5_ and all-cause mortality; others did not consider confounding factors that could have influenced the overall relative risk. For this reason, the results of this study regarding the premature deaths associated with PM_2.5_ may be somewhat uncertain. In addition, the study’s relative risks were derived from studies using land regression models and exposure assessments. We only estimated the change due to the stricter fuel restrictions from 1% to 0.1% in 2015 and could not analyze the effects of the earlier stepwise reductions. The populations for Sweden and the city of Gothenburg in both the high- and low-sulfur scenarios were used in the calculations with no adjustment for small population changes, which may have marginally affected the results. Further, the method of exposure modeling may have underestimated the exposure concentrations. The influence of meteorological conditions known to influence air pollution concentration levels in the atmosphere [64] was not accounted for and may also have influenced the estimation of adverse health effects. The study did, however, make estimations using empirical data and similar studies conducted in other coastal cities such as Stockholm and Malmö could contribute to our knowledge of the effects of pollution prevention measures.

## 6. Conclusions

The reduction in maximum sulfur content in shipping fuels led to decreased emissions and population exposures to PM_2.5_ from shipping in Sweden. Exposure estimations differed between the regional model and the city-scale model, with higher estimated population exposure in the latter, greatly influencing the health impact estimation. Interestingly, the spatial resolution of the population data did not have an impact on population-weighted PM_2.5_ exposure levels. Both the regional and the city-scale models showed a clear reduction after the implementation of the stricter 2015 SECA regulations on the Baltic Sea shipping emissions, leading to a decreased burden of disease due to outdoor PM_2.5_. Similar estimations may be made in the coming years to observe the effects of global sulfur fuel restrictions implemented in 2020 and other legislative changes. 

## Figures and Tables

**Figure 1 ijerph-17-07963-f001:**
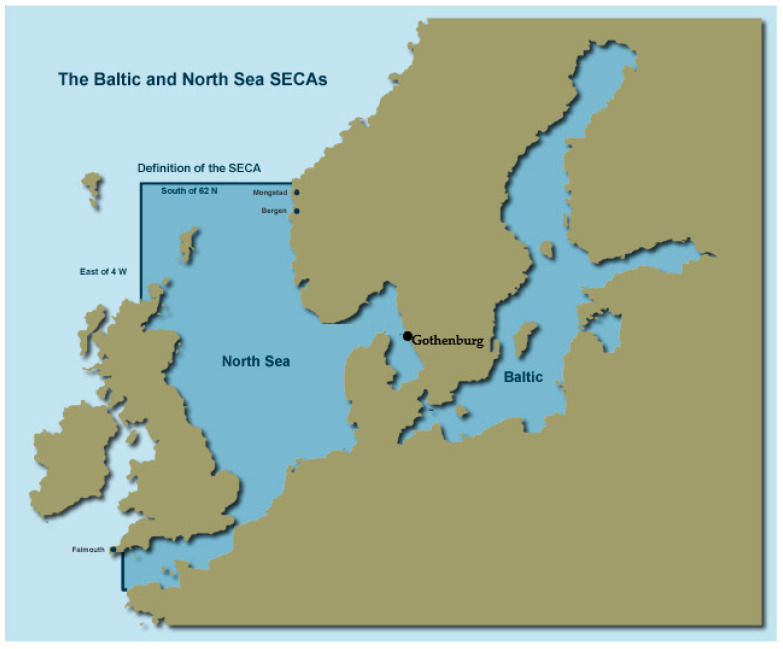
Sulfur emission control areas of the North and Baltic Seas.

**Figure 2 ijerph-17-07963-f002:**
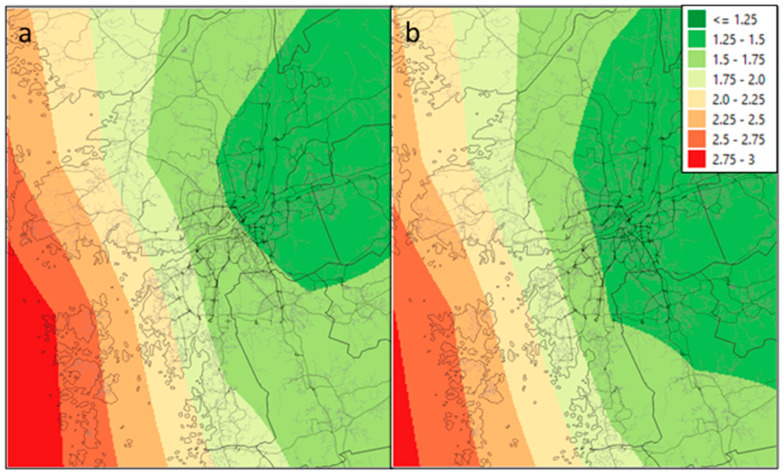
Exposure to PM_2.5_ (µg/m^3^) from shipping in the Baltic Sea in Gothenburg using a city-scale model: (**a**) high-sulfur; (**b**) low-sulfur.

**Figure 3 ijerph-17-07963-f003:**
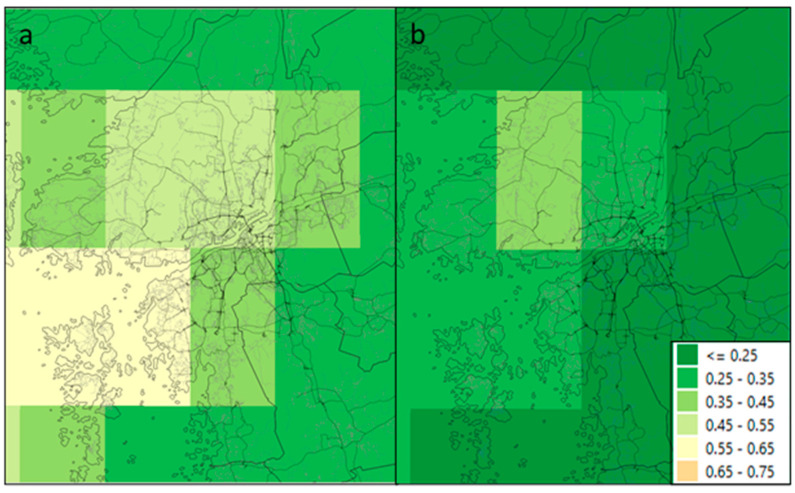
Exposure to PM_2.5_ (µg/m^3^) from shipping in the Baltic Sea in Gothenburg using a regional model: (**a**) high-sulfur; (**b**) low-sulfur.

**Figure 4 ijerph-17-07963-f004:**
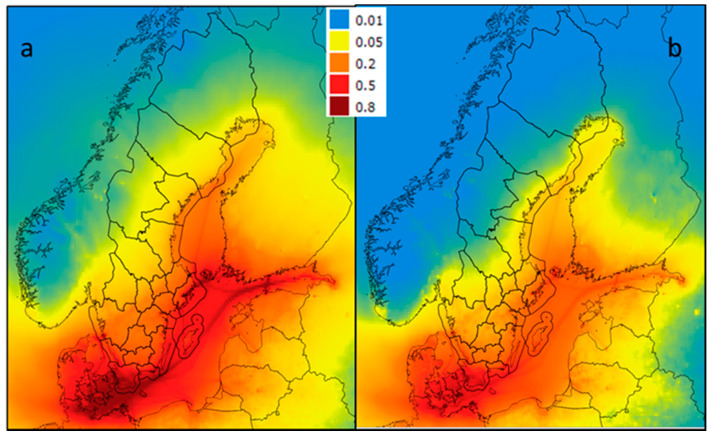
Comparison of the contribution from Baltic Sea shipping to PM_2.5_ (µg/m^3^) from ship emissions in Sweden and the surrounding area with the low-resolution EMEP model (**a**) high-sulfur; (**b**) low-sulfur.

**Table 1 ijerph-17-07963-t001:** Sweden’s 21 counties, populations in 2015, and population-weighted PM_2.5_ levels of exposure due to shipping using the regional model (1 × 1 km) grids, using both high- and low-sulfur scenarios.

County	Population	Average PM_2.5_ Population-Weighted Levels (µg/m^3^) High-Sulfur Scenario	Average PM_2.5_ Population-Weighted Levels (µg/m^3^) Low-Sulfur Scenario
Sweden	9,539,393	0.35	0.23
Stockholm	2,121,782	0.44	0.25
Uppsala	341,298	0.23	0.14
Södermanland	274,391	0.25	0.15
Östergötland	433,220	0.25	0.16
Jönköping	338,794	0.25	0.16
Kronoberg	185,656	0.33	0.22
Kalmar	233,322	0.39	0.26
Gotland	57,163	0.47	0.28
Blekinge	152,054	0.54	0.38
Skåne	1,259,881	0.73	0.54
Halland	304,989	0.46	0.33
Västra Götaland	1,615,664	0.32	0.21
Värmland	272,762	0.12	0.07
Örebro	282,761	0.17	0.10
Västmanland	255,707	0.19	0.12
Dalarna	276,209	0.07	0.06
Gävleborg	276,282	0.12	0.07
Västernorrland	241,726	0.11	0.06
Jämtland	126,088	0.04	0.02
Västerbotten	259,963	0.09	0.05
Norrbotten	248,378	0.06	0.03

**Table 2 ijerph-17-07963-t002:** Mean PM_2.5_ population exposure (µg/m^3^) from shipping emissions at city-scale and regional models using high- and low-sulfur scenarios for the Gothenburg area.

		City-Scale Model			Regional Model		
	Population (Pop)	High-Sulfur	Low-Sulfur	Difference	High-Sulfur	Low-Sulfur	Difference
Pop centroids	664,715	1.60	1.49	0.11	0.53	0.34	0.19
Pop 1 km grid	643,964	1.61	1.50	0.11	0.54	0.36	0.19
Pop 100 m grid	657,243	1.60	1.49	0.11	0.54	0.35	0.19

**Table 3 ijerph-17-07963-t003:** Estimated number of premature deaths due to PM_2.5_ emissions from Baltic shipping at the regional model in the high- and low-sulfur scenarios according to HRAPIE and ESCAPE exposure–response (ER) functions.

County	Mortality at Age ≥30 in 2015	Premature Deaths High-Sulfur	Years of Life Lost High-Sulfur	Premature Deaths Low-Sulfur	Years of Life Lost Low-Sulfur	Reduction (%)
Sweden	89,702	196–410	1820–4161	126–264	1172–2680	36
Stockholm	15,160	41–87	507–1159	23–49	283–648	44
Uppsala	2727	4–8	42–97	2–5	26–59	39
Södermanland	2884	4–9	37–84	3–6	22–51	39
Östergötland	4123	6–13	58–132	4–8	36–83	37
Jönköping	3352	5–11	46–106	3–7	30–68	35
Kronoberg	1823	4–8	33–75	2–5	22–51	32
Kalmar	2751	7–14	50–113	4–9	33–75	34
Gotland	617	2–4	14–33	1–2	9–20	39
Blekinge	1668	6–12	44–102	4–8	31–71	30
Skåne	11651	52–110	495–1132	39–81	366–836	26
Halland	2760	8–16	75–171	6–12	54–123	28
Västra Götaland	14769	29–61	281–642	19–40	182–417	35
Värmland	3215	2–5	17–39	1–3	10–24	38
Örebro	2905	3–6	25–58	2–4	16–36	37
Västmanland	2572	3–6	27–61	2–4	16–37	39
Dalarna	3135	1–3	10–22	1–2	8–19	16
Gävleborg	3313	3–5	18–42	1–3	10–24	44
Västernorrland	2988	2–4	14–31	1–2	7–16	48
Jämtland	1481	0.3–1	2–6	0.1–0.3	1.12–2.5	56
Västerbotten	2611	1–3	13–29	1–2	1–15	47
Norrbotten	2791	1–2	8–18	0.4–1	3–8	55

**Table 4 ijerph-17-07963-t004:** Estimated number of extra annual cases of myocardial infarction and stroke due to PM_2.5_ emissions from Baltic shipping at the regional model in the high- and low-sulfur scenarios.

County	Extra cases MI High-Sulfur	Extra Cases MI Low-Sulfur	Reduction (*n*)	Extra Cases Stroke High-Sulfur	Extra Cases Stroke Low-Sulfur	Reduction (*n*)	YLD MI	YLD Stroke	DALY High-Sulfur	DALY Low-Sulfur
Sweden	184	118	65	274	177	98	4	1559	5724	4243
Stockholm	36	20	16	60	34	27	1	266	1425	915
Uppsala	4	2	2	5	3	2	0	46	143	105
Södermanland	4	3	1.7	5	3	2	0	43	127	94
Östergötland	6	4	2	8	5	3	0	60	193	144
Jönköping	5	3	2	8	5	3	0	55	161	124
Kronoberg	4	3	1	5	3	2	0	29	104	80
Kalmar	5	3	2	7	5	3	0	35	148	110
Gotland	2	1	1	3	2	1	0	10	43	30
Blekinge	6	4	2	7	5	2	0	25	127	96
Skåne	49	36	13	76	56	20	1	193	1326	1030
Halland	9	6	3	10	7	3	0	41	212	164
Västra Götaland	27	17	9	44	28	15	1	253	895	670
Värmland	2	1	1	3	2	1	0	46	85	70
Örebro	2	2	1	4	2	1	0	41	99	78
Västmanland	3	2	1	4	3	2	0	41	102	79
Dalarna	2	1	0.2	2	2	0.3	0	60	83	80
Gävleborg	2	1	1	4	2	2	0	60	102	84
Västernorrland	2	1	1	2	1	1	0	42	74	59
Jämtland	0.3	0.1	0.1	0.4	0.2	0.2	0	22	28	25
Västerbotten	2	1	1	2	1	1	0	42	71	57
Norrbotten	1	0.5	0.6	2	1	0.8	0.2	49	67	57

YLD: years lost due to disability, DALY: disability-adjusted life years.

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
