# Peer review of "Health Impact of Air Pollution from Shipping in the Baltic Sea: Effects of Different Spatial Resolutions in Sweden"

_ijerph, 2020, doi:10.3390/ijerph17217963_

Round 1

Reviewer 1 Report

Title: Health Impact of Air Pollution from Shipping in the Baltic Sea: Effects of Different Spatial Resolutions in Sweden
Authors: Mwase et al.

Summary: The authors attribute particulate matter exposure to shipping and estimate exposure and health impacts. The methods follow typical counter-factual simulations with and without shipping controls, and compare two different simulations. A highlight of the manuscript is comparison of high-resolution and "low-resolution" modeling.

Response: The manuscript is generally of interest, but needs improvements before publication. The nomenclature throughout the paper is difficult to follow. There is no effort to establish credibility of the modeling. This is especially important because there is an intimation that higher resolution is better. The bulk of the manuscript spends enumerates health impacts and the differences between resolutions. Right now, we know that two models are different but no evidence of a controlled experiment is provided.

The difference between resolutions would be more valuable if resolutions were the only differences. The low-resolution model has barely a paragraph, but at least has a citation or two. The high resolution model has no citations and comparable description length to baseline population... Did the high resolution model use the same CTM? Did the high resolution model use the same meteorology model? Were other emissions comparable? If the models are extremely different, they should not be discussed a simply a test of resolution. A true test of resolution would be to "up-scale" (i.e., average up) the high resolution model and calculate the pop-weight average.

As written, the methods do not support the comparison of low and high resolution. The authors either need to supply supporting methods to demonstrate similarity of models or find another way to address this issue. The authors need to consider what high resolution would be in the context of empirically derived health relationships. A 100m resolution model of exposure is not consistent with a residence-distributed population. Finally, the health impacts sensitivity appears linearly proportional to population weighted concentration. They authors may want to acknowledge this and spend less of the manuscript enumerating them.

Line-by-line

* 25, "PM2.5 exposure from shipping in Gothenburg decreased 2014-2016 by 7%" -- Does this mean that the population weighted average for the three-year period would decrease by 7%? The rest of the paper has this problem too. Sometimes 2014 and 2016 are used as surrogates for before and after controls, but the paper suggests that all three years were simulated with and without... If so, the years should not be used as surrogates for with and without controls.

* 26, "low-resolution" is the coarse model 0.1 degree or approximately 10km resolution? If so, much work has been done showing that resolutions finer than this do not always perform better.

* 36, Outdoor, Indoor or both?

* 41, "levels" do you mean emissions? Or concentrations?

* 53-56, rephrase. Hard to follow.

* 56-57, are you separately discussing primary PM25 or does this include secondary PM25 from SO2?

* 62-63, authors need to specify what resolutions they consider "coarse." Does this include the AQMEII work?

* 64-67, Are the concentration changes in the linear-regime of the CRF functional forms? If so, what is the value of reporting more than the population weighted average? Would there be more value in establishing that finer resolution produced more "realistic" gradients?

* 79-82, Are there any published reports of the model that was run? Which model was it? What chemistry did it have? Has it been evaluated? Etc? Which meteorology years were simulated?

* 93, consider noting the county for people unfamiliar.

* 95, What is the source of population-weighted concentrations? Coarse or fine? Fine was only run for Gothenburg, right?

* 141, "of of" -> "of"

* 147, Are results using 2014 and 2016 or 3-year met with and without controls? Presenting the results with 2014 and 2016 makes this confusing. This is true throughout the paper. Also, the high resolution model methods seems to suggest only 2016 emission inventories were used, but do not discuss meteorology.

* 152 and 160, Were any observations available for comparison? Any satellite data? Anything to establish that either gradients were reasonable?

* 196, I spot checked Vastra Gotaland and Sweden. Table 3 appears to be in the linear region of the CRFs where % impact change is equal to % pop weighted change.

* 209, Similar to Table 3 -- appears to be in the linear range.

* 215, this line is an example of where years are used as a surrogate for emission year, and there is no mention of meteorological year(s).

Author Response

Dear reviewer

We thank the reviewers for valuable comments on how to improve our manuscript, and have revised the manuscript accordingly. In particular, we have extended and improved the sections on exposure modelling in methods, and added a paragraph to the discussion on exposure models. We have also in response to the reviewers comments rewritten the introduction to improve clarity, and the conclusions so they more precisely represent the findings of this study. An improved version of the manuscript is uploaded as both a “clean” version and one where changes are marked by “track changes”. Please find Attached the reviewers comments followed by individual responses point-to-point marked in red. We hope that with these changes you will find the paper acceptable for publication in International Journal of Environmental Research and Public Health.

Nandi Mwase

Reviewer 2 Report

This is an interesting research paper concerning Health Impact of Air Pollution from Shipping in the Baltic Sea: Effects of Different Spatial Resolutions in Sweden.

Major points to be reflected:

  1. The modeling period is from 2014 to 2016 with an new (stricter) regulation in 2015. This is not sound. The modeling period should include a longer time period.
  2. The paper deals mainly with the modeling approach. However as a calibration it is necessary to know some PM2.5 measuring data. Stockholm and some other cities might have the necessary measuring PM2.5 data at least to validate the results and to compare them with the two different models output and the differences between them.
  3. The conclusion chapter must reflect the results of the points explained in A and B.

Author Response

Dear Editors and Reviewers

Response to reviewers:

We thank the reviewers for valuable comments on how to improve our manuscript, and have revised the manuscript accordingly. In particular, we have extended and improved the sections on exposure modelling in methods, and added a paragraph to the discussion on exposure models. We have also in response to the reviewers comments rewritten the introduction to improve clarity, and the conclusions so they more precisely represent the findings of this study. An improved version of the manuscript is uploaded as both a “clean” version and one where changes are marked by “track changes”. Please find attached the reviewers comments followed by individual responses point-to-point marked in red. We hope that with these changes you will find the paper acceptable for publication in International Journal of Environmental Research and Public Health.

Nandi Mwase

Round 2

Reviewer 1 Report

Title: Health Impact of Air Pollution from Shipping in the Baltic Sea: Effects of Different Spatial Resolutions in Sweden
Authors: Mwase et al.

I think shipping emissions is an important topic, but this paper needs improved methods, results, and discussion in order to be publishable. The methods need more clarity and the emissions, in particular, have almost no characterization at all. The figures are poorly captioned, missing units, and color scales are not sufficiently descriptive. The discussion make speculative remarks about basic parts of their own emission inventory. In its current form, this is not publishable. I recommend that the authors revise.

Revision Response:
* The authors have done a good job updating details of the city-scale model.
* The updated description of the city-scale modeling is still somewhat imprecise.
* Airviro has multiple model options (See below) and no clarity is provided on which one was used.
* What meteorology was used to drive which ever model was used?
* Which ever gaussian model and meteorology they use, the gaussian results are added to the regional results. Were the regional results also both with and without enforcement of the ECA? Or a regional model without shipping? If with shipipng, could that help to explain the higher concentrations?
* The authors have done an inconsistent job updating nomenclature (pre-SECA (2014), post-SECA (2016)), which is now clearly not appropriate for the city-scale results.

Line-by-line:

* 59, Figure 1 is of insufficient quality to read what I can only assume is Gothenburg or any of the other city names and what may be a coordinate. Not sure, can't read it.

* 79, The use of "local" here is ambiguous. It could mean both Sweden and the broader region. From the description of the city-scale, it is not obvious that STEAM was used in the city-scale simulation. If it was, make that clear like you did for EMEP.

* 79, Also, STEAM outputs are critical part of your study. What was the mass of sulfur emitted from ships in your regional domain? In your city-scale domain? In the regional domain overlapping your city scale domain? A kgS/yr estimate would be useful.

* 95, linking to a website is an insufficient citation. Has this model ever been evaluated in the peer-reviewed literature? The website says that "Airviro supplies both its own unique models and many of the most established open models in the field of AQM (see list below), such as the AERMOD and CALPUFF models recommended by the U.S. Environmental Protection Agency (US/EPA) and the AUSTAL2000 model developed on behalf of the German Federal Environmental Agency (UBA)." It would seem that it could be an Airviro-specific model, AERMOD, or CALPUFF.

* 99, by local sources do you mean all non-shipping?

* 101, Adding guassian concentrations to "background" from a Eulerian model is not uncommon, but it does create double counting. In this case, there is no discussion of how that was or was not treated or addressed. For example, were shipping emissions removed from the Eulerian model before simulation and combination with the Gaussian results?

* 102, The text is insufficient and unsupported by the citation. "The results from this procedure corresponded very well with PM2.5 measurements made by the local environmental office in Gothenburg [38], without any empirical adjustment of the model results being made." I went to the report linked by [38]. Using Google Translate and skimming the report, I saw no evidence of "correspond[ing] very well with PM2.5 measurements." In fact, I saw no measurements in that report at all.

* 172, Is Figure 2 the entire city scale model domain? If so, it would appear that the edge of teh modeling domain is very close to a shipping lane.

* 193, In Figure 4 and S2, there are no units on the figure. In Figure 4, what are the bin minimums and maximums? Am I right in interpreting that there are 0.01 units? of shipping "emissions"? Or are these concentrations? Figure 4 is also not of Gothenburg, but of the region. All figures should have units. All figures should have clear color scales. All figures should be clearly and obviously labeled as either emissions or concentrations. According to Figure S2, there were larger shipping emission decreases in the city-scale model... Or are these concentrations? If emissions, I suggest masking out cells with 0 emissions to help the reader understand the emissions. If these figures are not emissions, but are concentrations, then you need to provide some maps of emissions.

* 280-284, This suggests that the pre-SECA shipping emissions were not the same. If the fuels being assumed in the city-scale model are different than those in the regional model, then this is not a study about resolution. Second, you developed the emissions with STEAM. You should be telling us why they are different rather than speculating what might have happened in your work.

* 286, "The study" if so, which one? Or "This study" meaning your study? if so, I don't see that.

* 289, "This study shows marked improvements"... "This study" should be reserved for discussing your own results. I think you are using it to reference the citation in the previous sentence, which is very confusing when the previous sentence made a statement and the citation was not directly referenced. It is even more confusing when the previous citation referenced two studies (e.g., line 286).

* 305, in "2016" or in the "low-Sulfur scenario"

* tables, I suggest adding Gothenburg (city-scale) and Gothenburg (regional scale) to the tables. Right now, the Gothenburg results are hard to find and inconsistently discussed with respect to the low-Sulfur scenario.

Author Response

We thank the reviewer again for their valuable comments on how to improve our manuscript and have revised the manuscript accordingly. In particular, we have improved the sections on exposure modeling in methods as well as the supplementary materials. We have also in response to the reviewer's comments rewritten the highlighted issues made in the reviewers’ comments to provide clarity and improved the images used in the manuscript. An improved version of the manuscript is uploaded as both a “clean” version and one where changes are marked by “track changes”. In the attachment are the reviewers’ comments followed by individual responses point-to-point marked in bold and highlighted in red. We hope that with these changes you will find the paper acceptable for publication in International Journal of Environmental Research and Public Health.

Nandi Mwase

Reviewer 2 Report

The authors did pay attention to all the points raised by the reviewer and the paper has been rewritten accordingly.

Author Response

Dear Editors and reviewer

We are glad that we have adequately addressed the comments made earlier. We have made some improvements to the paper based on the other reviewers’ comments. An improved version of the manuscript is uploaded as both a “clean” version and one where changes are marked by “track changes”. We hope that with these changes you will find the paper acceptable for publication in International Journal of Environmental Research and Public Health.

Nandi Mwase